# Reorienting Nurturing Care for Early Childhood Development during the COVID-19 Pandemic in Kenya: A Review

**DOI:** 10.3390/ijerph17197028

**Published:** 2020-09-25

**Authors:** Constance Shumba, Rose Maina, Gladys Mbuthia, Rachel Kimani, Stella Mbugua, Sweta Shah, Amina Abubakar, Stanley Luchters, Sheila Shaibu, Eunice Ndirangu

**Affiliations:** 1School of Nursing and Midwifery, Aga Khan University, Nairobi 00623, Kenya; maina.rose@aku.edu (R.M.); gladys.mbuthia@aku.edu (G.M.); rachel.kimani@aku.edu (R.K.); sheila.shaibu@aku.edu (S.S.); eunice.ndirangu@aku.edu (E.N.); 2Department of Population Health, Aga Khan University, Nairobi 00100, Kenya; stanley.luchters@aku.edu; 3Africa Early Childhood Network, Nairobi 00502, Kenya; stellashiku2005@yahoo.com; 4Global Programs Team, Aga Khan Foundation, 1211 Geneva, Switzerland; sweta.shah@akdn.org; 5Institute for Human Development, Aga Khan University, Nairobi 00100, Kenya; amina.abubakar@aku.edu; 6International Centre for Reproductive Health, Department of Public Health and Primary Care, Ghent University, 9000 Ghent, Belgium; 7Department of Epidemiology and Preventive Medicine, Monash University, Victoria 3800, Australia; 8Burnet Institute, Melbourne 3004, Australia

**Keywords:** COVID-19, impacts, nurturing care, early childhood development (ECD), maternal, newborn, child health, child growth development, early brain development, vulnerable children and families

## Abstract

In Kenya, millions of children have limited access to nurturing care. With the Coronavirus disease 2019 (COVID-19) pandemic, it is anticipated that vulnerable children will bear the biggest brunt of the direct and indirect impacts of the pandemic. This review aimed to deepen understanding of the effects of COVID-19 on nurturing care from conception to four years of age, a period where the care of children is often delivered through caregivers or other informal platforms. The review has drawn upon the empirical evidence from previous pandemics and epidemics, and anecdotal and emerging evidence from the ongoing COVID-19 crisis. Multifactorial impacts fall into five key domains: direct health; health and nutrition systems; economic protection; social and child protection; and child development and early learning. The review proposes program and policy strategies to guide the reorientation of nurturing care, prevent the detrimental effects associated with deteriorating nurturing care environments, and support the optimal development of the youngest and most vulnerable children. These include the provision of cash transfers and essential supplies for vulnerable households and strengthening of community-based platforms for nurturing care. Further research on COVID-19 and the ability of children’s ecology to provide nurturing care is needed, as is further testing of new ideas.

## 1. Introduction

The Coronavirus disease 2019 (COVID-19) pandemic is spreading in unprecedented ways and has a significant impact on nurturing care and early childhood development outcomes. Currently, there are over eighteen million COVID-19 confirmed cases globally, Kenya recording over 20,000 cases as of 4 August 2020. Kenya reported her first case of COVID-19 on 13 March 2020 [1]. On 15 March 2020, the government ordered a lockdown, including the closure of all schools within the Republic of Kenya. While these restrictions, such as closures of educational institutions, stay at home directives and cessation of social gatherings, have the potential to curb the spread of the infection, they have been detrimental to the very fabric that defines the social interaction norms in the Kenyan context. The COVID-19 pandemic and the associated responses have posed unique challenges to all sectors, including those implementing nurturing care for early childhood development (NCfECD) [2].

This review paper looks at the current impacts of COVID-19 on nurturing care in Kenya, with relevance to the sub-Saharan African region, focusing on the period of conception to 4 years of age. The care of these children is often delivered through caregivers or other informal platforms as there is no guiding national policy for children below four years. Through a conceptual framework, the objective of this paper is to elucidate how nurturing care for the youngest Kenyans is being threatened. We also present our reflections on how to reorient and support nurturing care during and after the pandemic period. Furthermore, we describe a myriad of measures and strategies that key stakeholders in Kenya can adopt to reduce these threats to young children’s ability to survive, thrive and transform their societies.

## 2. Nurturing Care Framework (NCF)

Nurturing care refers to environments created by parents or caregivers and public policies, programs and services, that guarantee children’s health, adequate nutrition, safety and security (protection), opportunities for early learning all provided by responsive caregivers [3]. In May 2018, WHO member states adopted the nurturing care framework (NCF), which provides the evidence-based blueprint to support the attainment of holistic growth and development of children through inter-sectoral collaboration [3]. Although early childhood development (ECD) covers children aged 0–8 years, the NCF centers on the foundational period from pregnancy to three (3) years [3]. Provision of nurturing care during this period is vital as the science shows that the period from pregnancy to age 3 is the most critical for brain development [3]. Approximately 80% of a baby’s brain formation occurs during this window [4]. 

The NCF is situated within an ecological framework of an enabling environment which includes the following: caregivers’ capabilities; empowered communities; supportive services; and enabling policies [3]. It is well-recognized that an adverse environment impairs ECD, with both short- and long-term impacts. In low- and middle-income countries (LMICs), close to 250 million children below five years were already at risk of sub-optimal development prior to the COVID-19 pandemic due to extreme poverty [5].

## 3. Kenyan Context

In Kenya, the proportion of the population living below the poverty datum line of USD 1.90 per day [6] was 36% with an estimated 50% of the urban population, residing in informal settlements [7]. An estimated one million (30%) of Nairobi’s children live in informal urban settlements with poor infrastructure and limited access to education and health services as well as nurturing care [3,7]. The prevalence of stunting in informal urban settlements stands at 26.3% [8] which is similar to the national prevalence of 26.2%. Further, the HIV prevalence among children in Kenya is estimated at 7% [9]. Children from the most vulnerable groups, such as those living in informal urban settlements and affected by HIV and AIDS, are also most likely to lack consistent stimulation, proper nutrition and nurturing care. 

Public sector childcare or early learning services are limited for children below 4 years. Further, there is no policy and legal framework to guide the services provided by the private sector for these children and their families. With the COVID-19 pandemic, it is anticipated that children will bear the biggest brunt of the direct and indirect impacts of the pandemic, especially those in LMICs such as Kenya where many children are already at risk of not achieving their full potential. Therefore, there is a need to mitigate the impact of COVID-19 by prioritizing programs and policies that support the continuum of ECD [10].

Anecdotal evidence in Kenya shows that the COVID-19 pandemic is contributing to deteriorating optimal environments that threaten children’s early development and has direct health impacts on caregivers and children [11,12,13]. Strategies are required to prioritize a range of ECD interventions during the COVID-19 pandemic to support caregivers so that they can meet the needs of their young children. ECD goes beyond improving child survival to enabling children to reach their full potential through cognitive, socio-emotional and physical development. Failure to prioritize NCfECD will lead to a future pandemic where children who are presently most vulnerable, will have significant deficits on their health, wellbeing and productivity.

## 4. Evidence from Previous Pandemics

Previous pandemics have had long-term negative impacts over multiple generations. The development of children who were exposed to the Asian influenza pandemic in 1957, while in utero, was hampered with evidence of poor cognitive development [14]. The 1918 Spanish flu was reported to lower educational attainment for those individuals whose mothers had potential in utero exposure [15]. In Japan, primary school children born between 1919 to 1929 were shorter than those in surrounding cohorts [16]. The timing of the prenatal exposure to influenza was also reported to have had worse consequences in those who were exposed in early gestational 0 to 8 weeks, as it was associated with delayed psychomotor development at 6 months of age [17]. In fact, it has been established that the Spanish flu had negative outcomes in later life for those who were exposed in utero in several countries such as the USA [18], Brazil [19], Switzerland [20], and Taiwan [21]. In a narrative review of infants and children with congenital Zika virus, epilepsy and motor abnormalities were noted [22]. Wearing masks to prevent the transmission of SARS also negatively impacted communication between children and adults, and was also threatening to children who had been sexually abused [23]. The HIV pandemic had a negative impact on child growth and development. Globally and in sub-Saharan Africa including in Kenya, young children affected by HIV particularly those who are HIV-infected, have a high risk of mental health problems, neurocognitive deficiencies, developmental delay, and poor nutrition outcomes [24,25,26,27,28]. The HIV pandemic generated a lot of lessons related to ECD. However, for a very long time, ECD was associated with child survival only, without a strong focus on promoting thriving and transforming [29]. 

ECD-related outcomes such as cognitive impairment and developmental delays as well as long-term impacts across the life course have not been tracked widely. Even where there are attempts to focus on thriving and transforming, during epidemics and pandemics, the focus reverts to child survival. Despite the frequency of epidemics, there is a scarcity of research on holistic ECD outcomes in the sub-Saharan Africa context. Where research in sub-Saharan Africa exists, it has focused on other types of emergencies such as conflict and refugee crises [30].

## 5. Impacts of COVID-19 on Nurturing Care

### 5.1. Conceptual Framework of the Impacts of COVID-19 and Control Measures on Nurturing Care

There are several far-reaching, interlinked direct and indirect impacts of the COVID-19 pandemic and associated control measures on nurturing care and related ECD outcomes including children’s cognitive, physical, language, motor, and social and emotional development (Figure 1). These include direct physical and mental health impacts resulting in illness and/or death from COVID-19 infection, and severe stress leading to deterioration of mental health and well-being. With deaths increasing, many children are becoming orphaned or experiencing greater adversity. COVID-19 has also affected access to health and nutrition systems including routine immunizations. Social impacts include increased teenage pregnancies and a rise in gender based violence, all with a bearing on ECD outcomes. They also include lack of social and child protection services to support parents and primary caregivers. Furthermore, child development and learning impacts including lack of access to institutional-based childcare services and critical nurturing environments have become more severe during this period, affecting the learning that children need during the most critical period of brain development. Finally, the economic impacts have a direct interplay with all other impacts, and have contributed to increased stress among caregivers and children. In some places, it has resulted in food insecurity, thereby also influencing children’s physical health. All these impacts have short-term consequences that will translate into long-term changes in children’s life trajectories.

### 5.2. Direct Health Impacts of COVID-19

#### 5.2.1. Impacts on Children’s Health

COVID-19 is an evolving pandemic, and despite the worldwide spread, the effects of COVID-19 on pregnancy, childbirth in addition to newborns and toddlers are not well-established, and the evidence is mixed. Recent experience suggests there is a low risk of intrauterine infection by vertical transmission in women with COVID-19 disease [31,32,33], although the first case of vertical transmission was recorded in July 2020 in India [34]. A systematic review by Zimmermann and Curtis (2020) [35] on COVID-19 in children, pregnancy and neonates reported fetal distress in 30% of pregnancies, with 37% of women having preterm deliveries. Neonatal complications including respiratory distress or pneumonia (18%), disseminated intravascular coagulation (3%), asphyxia (2%) and two perinatal deaths were also reported.

The epidemiological and pathophysiology of COVID-19 in children remains unclear. Evidence from China [36], Italy [37], Netherlands and the UK [38] indicate that children represent less than 5% of diagnosed COVID-19 cases. However, children under one year and those with comorbidities such as asthma are more likely to be hospitalized [39]. Although COVID-19 in children seems to have mild symptoms, there is a high prevalence of pneumonia associated with COVID-19 in children at 53% [40]. The majority of children have less severe symptoms, and thus are less likely to be tested, leading to an underestimate of child infections. Indeed, studies confirm that severe illness and mortality from COVID-19 is rare in children [41]. However, there are emerging concerns of a novel severe Kawasaki-like disease in children related to COVID-19 that may represent a post-COVID infectious syndrome [38]. In a systematic review, children were found less likely to be the main drivers of the pandemic compared to adults who get severe disease [41]. In Kenya, the reported cases of young children testing positive for COVID-19 are low; 9% of reported cases as of 27th July were children aged 0–9 years with a 2% case fatality rate [42].

Although children do not account for the majority of cases, they are likely to face the most substantial impact of the COVID-19 pandemic [43]. Consequently, a focus on children, and especially the youngest, is vital not only due to the impact that they may face during the current crisis, but also because the negative impact has the potential of persisting across their lives in many years to come. 

#### 5.2.2. Psychosocial Impacts on Caregivers and Children

There is also heightened stress and psychosocial difficulties among parents and caregivers that threaten the provision of optimal nurturing care environments which children need to achieve their potential [12]. Worry, stress, and being anxious have been reported among 75% of Kenyans due to COVID-19 [44]. School and daycare closure, job losses, economic uncertainty, inability to afford food and lack of access to essential services have resulted in increased stress and anxiety among caregivers. Some level of stress is normal and can even be beneficial when it is positive, but when it is elevated, constant and adds to existing adverse conditions, it can become toxic [45]. Toxic stress can have long-term impacts on a person’s hormones, thereby affecting a child’s brain architecture, physiological and chemical makeup, and overall development over a lifetime that may never be fully reversed [45].

The Ministry of Health in Kenya developed a comprehensive guide for health workers on mental health and psychosocial support during the COVID-19 pandemic to cover the needs of the population and people on treatment for COVID-19 [46]. However, it is not clear if this has been widely disseminated, and the extent to which various population segments are benefitting from the guidance is also unclear. Access to material and psychosocial support, caregivers’ and families’ ability to cope with the pandemic and its consequences may be limited, and they may not be able to provide effective nurturing care [10,47]. When children experience trauma, experience adversity and lose secure attachment and bonding due to deficiencies in responsive caregiving, they experience stress, which has negative impacts on their health, wellbeing and lifelong learning, including a higher risk for developing a variety of cognitive, behavioral and emotional difficulties later in life [45].

### 5.3. Indirect Health Impacts

#### Impact on Health and Nutrition Systems

Aside from the direct health impacts on the physical and mental health and wellbeing of children, caregivers and families, COVID-19 has also affected health systems and disrupted access to routine nurturing care services. COVID-19 has placed a strain on the overstretched healthcare systems, a key entry point for nurturing care, and disrupted the delivery of vaccination of children under five years due to supply chain and human resource constraints [48]. The weak health systems in LMICs such as Kenya are vulnerable to the spread and impact of COVID-19, having witnessed service disruptions and lack of preparedness in the face of the crisis. The basic tenets of the right to health are being tested. Public health expenditure as a percentage to GDP is deficient in the region, and Kenya stands at 5.7%, far below the recommended 15% per the Abuja Declaration [49]. The country already had a shortage and maldistribution of health workers, but with COVID-19, the disparities in access to healthcare between the rich and the poor in urban areas, as well as between the rural and urban divide, are widening.

Though most nurturing care interventions in Kenya begin at birth, maternal preconception health and wellbeing influences child development. Intrauterine growth restriction has been linked to adverse outcomes including prematurity, low birth weight, stunting, anemia, neurodevelopmental conditions, stillbirths and child mortality [50,51]. Evidence from LMICs shows that reproductive, maternal, newborn and child health interventions including iodine, iron and folate supplementation during preconception have had a significant impact on children’s cognitive, physical and socio-emotional wellbeing [52]. The uptake of preconception care in Kenya is very low since over 40% of the pregnancies in the country are unintended [53]. Notably, the majority of the unintended pregnancies occur in young girls who take time to acknowledge their pregnancies. This leads to delay of the first antenatal visit and, in some instances, non-uptake of antenatal services throughout the pregnancy [54]. In a context where COVID-19 has led to an increase in the number of teenage pregnancies as well as a disruption in routine care, the net effect will be delays and low uptake of antenatal services. Consequently, there may be an increased risk of infant and maternal morbidity and mortality. The COVID-19 pandemic has disrupted maternal and child health and nutrition, including antenatal, skilled delivery and postnatal services, as well as immunizations, health education and promotion, all resulting in a reversal of the previous gains made in reducing maternal and neonatal mortality [55].

The reduced accessibility of essential maternal and child health and nutrition services is worsening ECD outcomes and further exacerbating disparities among vulnerable households such as those living in informal urban settlements. The pandemic threatens the continuity of critical and essential services for expectant women, newborns, and children under five years including those with disabilities and developmental delays. The reluctance of parents to visit clinics due to fear of infection with COVID-19 may also interrupt immunization and other child health programs [56]. The social distancing, lockdown and curfew measures have led to decreased utilization of maternal health services. Pregnant women experience challenges in accessing health and nutrition services, which has been worsened by the COVID-19 crisis. For example, lack of transport during lockdown and curfews, and fear of visiting health facilities due to concerns regarding COVID-19 infection have been observed [57]. Maternal and child malnutrition, including micronutrient deficiencies and child stunting are expected to increase [58].

Mothers and children need access to key essential nutrition actions and services so that they are well-nourished pre-conceptually, intrapartum and during lactation. Furthermore, they also need services to diagnose and address micronutrient deficiencies through iron and folic acid (IFA) supplementation to prevent neuro-developmental disabilities in children [59]. However, anecdotal reports confirm reduced utilization of maternal and child health services in Kenya, worsened by infection of some health workers leading to the suspension of maternal services [60]. Likewise, in Sierra Leone and Liberia, the Ebola crisis exacerbated the poor health outcomes within weak health systems [61,62]. Liberia and Guinea experienced a sharp decline of more than 25% in the monthly number of children vaccinated against measles in 2014 and 2015 due to the Ebola outbreak as compared to the previous years [63]. The indirect effects of Ebola on maternal and child health were believed to be greater than the direct consequences [62]. Antenatal care, family planning, facility delivery and postnatal care were adversely affected, leading to an increase in maternal neonatal and stillbirth deaths in 2014–2015 [61].

### 5.4. Economic Impacts

#### Losses in Income and Increased Poverty Levels

The economic well-being of a family affects a child’s ECD outcomes because it affects the child’s ability to be in a safe and protective home and access health services and programs and nutritious foods, all of which cost money. Children growing up in vulnerable households face even greater challenges to thrive given the pandemic and existing adversities [64]. The directives to reduce transmission through social distancing, hand-washing, self-isolation and self-quarantine for 14 days for those exposed to the virus may be unattainable for informal settlement residents who have space limitations and limited access to water, sanitizers and masks. Families who were already vulnerable prior to the pandemic have been pushed to dire circumstances with losses in income and are unable to afford basic necessities, while others juggle work and childcare among other responsibilities. Stay at home orders and lockdowns are unlikely to be followed through as quest for food and basic commodities is necessary [65]. 

The economic impacts of the pandemic are anticipated to have far-reaching consequences on long-term health and wellbeing of the population compared to the direct health impacts [66]. There is a downward trend in the Kenyan economy marked by job losses, inconsistent food supply and an increase in stress levels among adults and children [67]. The pandemic has caused a severe unemployment crisis in Kenya, with at least one million people having lost their jobs or been placed on indefinite unpaid as of June in both the formal and informal sectors [68]. There was a marked decline in labor force participation from 75% in 2019 to 57% in April 2020, and women are the most affected with a participation rate of 49% compared to men at 65.3% [69]. The government has introduced various fiscal policy and income support measures such as tax waivers, reduction in taxes for all micro, small and medium enterprises, as well as COVID-19 emergency funds and earmarked funds for social protection in the form of cash transfers [66]. However, the number of vulnerable families continues to increase as the pandemic persists.

Families living in informal settlements live in overcrowded areas and lack basic housing, water and sanitation, which make them vulnerable to disease outbreaks despite having the knowledge of COVID-19 measures [65,70]. Contact tracing has shown local transmission of COVID-19 to rise as community transmission becomes a significant driver, especially with people living in a big families leading to an increase in deaths [71]. This can be related to the respiratory viral transmission of COVID-19 through direct contact in the households where space is inadequate and social distancing impossible. Families also continue to experience other non-COVID-19-related health challenges coupled with movement restrictions, placing caregivers of children at greater risk of morbidity and mortality. As the situation continues to unfold and countries adopt this ‘new normal’, the potential negative impact of the prevailing situation on unborn and young children cannot be ignored [57]. Holistic child development requires a stimulating, safe environment, social interaction, education opportunities and adequate nutrition, all of which have been affected in one way or another [57]. The resultant economic impacts of COVID-19 have been felt at household level with a ripple of negative impacts on nurturing care.

### 5.5. Social and Child Protection Impacts

#### 5.5.1. Increase in Abuse, Neglect, and Violence

There has also been increased risk of abuse, neglect and violence against children of all ages [72] and domestic violence in Kenya [73,74]. Children with developmental delays and disabilities are very vulnerable and are often subjected to stigma and various forms of neglect and abuse [75]. Stress and anxiety among children are also likely due to disrupted routines. Routines are critical to enabling children to thrive in supportive environments in the home, childcare and early learning centers. All these circumstances mean that children in LMICs such as Kenya are at risk of faltering outcomes, as caregivers find it challenging to provide their children with the nurturing care they need during this pandemic. Children need a safe, secure and loving environment, yet these stressful experiences in early life increase the risk of developmental delays and non-communicable disease in later life [3,10]. Therefore, to promote safety and security, families and children need to live in safe environments, where children experience supportive discipline and do not experience neglect or violence. Responsive caregiving ensures sensitivity to children’s cues, thus promoting play and stimulation for early learning through day-to-day activities as well as caregiver–child interactions that are enjoyable [76,77].

It is plausible that with the lockdown and restrictions on movement, caregivers and families may have limited access to child protection services and programs. Where the services are present, they may experience difficulties reaching and providing care to vulnerable children. The closure of “babycares” may have implications for child protection, as the children are not being looked after by caregivers who offer an environment with some level of safety and security. Due to the ongoing crisis, children may also be locked up and restricted from exploring their environments or playing with other children due to fear of infection. Without access to social protection, caregivers facing heightened vulnerability due to loss of income may lack the safety nets to provide for and protect children.

#### 5.5.2. Orphanhood

In cases where caregivers succumb to direct COVID-19 infection or due to the indirect health impacts of COVID-19, children are orphaned. This affects children’s access to basic needs and nurturing care [78]. The experience of bereavement itself is a form of adversity, and could lead to emotional and psychological trauma, and induce fear and a sense of helplessness in children without positive coping mechanisms [79]. Orphaned and vulnerable children have an increased risk of being neglected, harmed, exploited, and they may experience gender-based violence, including early marriage. They also miss out on opportunities for play, a crucial aspect of child development and early learning. In Kenya, there is weak oversight of services to support orphaned and vulnerable children. Traditionally children would live with other relatives, and, in general, family-based care is preferred to institutional care where there are reports of abuse, neglect and exploitation [80]. However, with COVID-19 putting increasing food insecurity and economic hardships of families, orphaned and vulnerable children may not be supported in these families without the provision of safety nets.

#### 5.5.3. Teenage Pregnancies in the Pandemic and the Implications for Nurturing Care

High teenage pregnancy is not new in Kenya. Data from the Demographic and Health Surveys show that almost 2 out of 10 girls between the ages of 15 and 19 are reported to be pregnant or already had a child [81,82]. This trend has been fairly consistent for more than two decades with little change in prevalence between 1993 and 2014. Nevertheless, in light of the COVID-19 pandemic, the trend of teenage pregnancy is already showing signs of being more severe as a result of prolonged school closure, sexual violence and the declining economic situation in Kenya [83]. This trend is dire, as girls from poor families across the country are engaging in transactional sex to acquire money to buy sanitary pads and food [83,84]. Globally, it is predicted that due to the harsh economic times, the number of girls involved in survival sex will increase [85]. Previously, girls were able to access free sanitary towels through their schools; however, this is no longer the case since schools were closed following the COVID-19 crisis [86].

Teenage pregnancy presents significant health consequences to both mothers and newborns. Complications in pregnancy and childbirth are the leading cause of death among girls aged 15–19 years globally [87]. The risks are even higher for girls below the age of 16 years. Pregnant adolescents face a higher risk of eclampsia, endometritis and puerperal infections than women aged 20–24 years [88]. In addition, adolescent births are more likely to result in preterm births, low birth weight and newborns with severe congenital conditions. Furthermore, teenage pregnancy is a major contributor to a never ending cycle of ill-health and poverty [87].

The impact of teenage pregnancy includes loss of education opportunities, early marriages, and economic disempowerment [83,89]. Studies have shown that most teenage pregnancies occur among teenagers from deprived backgrounds [81,90]. Therefore, all these factors result in the intergenerational transmission of poverty from the teen mothers to their children with poor ECD outcomes. The situation is bound to get worse with the COVID-19 pandemic. Furthermore, cases of gender-based violence, in particular, child and early marriages are also on the rise [83,91,92]. It is well understood that children of teenage mothers tend to have poor ECD outcomes. The children have lower IQs and academic achievement, and are at a greater risk of repeating a grade. They are also at a greater risk of perinatal death and having a fatal accident before turning one year old [93,94]. The Ministry of Education announced that all schools within the territory of the Republic of Kenya shall remain closed until January 2021 [95]. This announcement is worrisome given the increasing cases of teenage pregnancy during the extended period of schools’ closure [96].

### 5.6. Child Development and Learning Impacts

The closure of daycares and pre-primary classes, which includes children up to four years, has affected children’s access to early learning, that is, building their brains in a safe and stimulating environment and developing their social and emotional skills while their parents work. Children learn best through play and interaction with peers; with daycares and other early learning centers closed, many children are not able to receive these critical inputs. These centers are also important sites for immunizations, meals and psychosocial support, all of which are being disrupted due to COVID-19 [97].

Prior to the COVID-19 pandemic in Kenya, along with the rising urban population and the need for parents to find informal work, there was a growing demand for childcare and early learning services. High unemployment and literacy rates of parents, and the absence of extended family support and public amenities and the prohibitive cost of quality childcare services led families relying on informal childcare centers as they sought employment. There was a proliferation of relatively low-cost, non-regulated and informal privately owned childcare centers for children aged three years and below, commonly referred to as ‘babycares’ with at least 2700 of them in Nairobi [98]. These informal babycares are often home-based or faith-based and lack the minimum standards, expertise and infrastructure required to support children to attain their developmental potentials. Some of these have poor lighting, are crowded with children that sleep most of the time, lacking play and stimulation and being served nutrient-poor or deficient foods. This large number is exacerbated by the lack of policy and legal framework to guide the services they provide for children below four years and their families. The 2006 National ECD Policy Framework of 2006 was not implemented due to operational issues [99].

Numerous conversations among stakeholders continue on the state of ECD for children below the age of four. With the nurturing care framework adopted in Kenya, the focus on children below four years is taking center stage alongside the prioritization of programs and services to meet their needs. While nascent, at the beginning of COVID-19, there had been considerable traction. With the pandemic, these gains are threatened, as policy makers’ focus and funds have been diverted to physical health, which includes preventing and treating those with COVID-19, rather than considering all aspects of child development. For middle- and upper-class families, they can hire childcare or early learning support, but this is out of reach for poor families.

## 6. Policy and Program Strategies to Reorient Nurturing Care

During the COVID-19 pandemic and beyond, the Kenyan government and other ECD stakeholders interested in ensuring that the youngest of children in the country are able to survive, thrive and continue on a positive life trajectory can reorient nurturing care. This is possible through utilizing the lens of direct health; health and nutrition systems; economic and social protection; and child development and early learning. Children’s needs are inter-related and holistic and so support must also possess these qualities. Kenya has a number of policies and systems in place to bolster nurturing care during the COVID-19 pandemic, but as is the case with many countries around the world, they are not fully financed and operational.

### 6.1. Direct Health and Health and Nutrition System Support

Actions to mitigate the negative impact on maternal, newborn, child and adolescent health need to be addressed by borrowing, developing, and implementing strategies utilized in previous epidemics and pandemics. This will guarantee continuity of care and avoid a rise in maternal and newborn morbidity and mortality. Support to caregivers and families would enable them to nurture their young through a multi-sectoral approach that builds on existing programs [100].

It is crucial to examine existing evidence on the direct effects of COVID-19 on maternal and newborn care and develop programs that target easy access to maternal and newborn health services, warranting safety for mothers, children and health professionals following the guidelines. This could include increased bottom-up community health education and promotion strategies on the current COVID-19 guidelines, utilizing a multi-sectoral approach through establishment of partnerships with community gatekeepers to teach mothers and caregivers. These strategies should be designed to be evidence-based and culturally appropriate, leading to holistic well-being for caregivers. Particularly, families and caregivers of children with developmental delays and disabilities require targeted support that meets their needs during the ongoing crisis, enabling them to practice responsive caregiving through ensuring child safety and security [47]. These children and those who are orphaned should be prioritized for social protection interventions implemented by both the government and development agencies. This support could include some or a combination of the following: cash transfers, food packs, mobile health and nutrition services, as well as regular support and monitoring by child protection teams. Children with disabilities may experience stress, have underlying health conditions that increase their risk of complications from COVID-19, and may also be unable to access therapy during this period. Similarly, their caregivers may have heightened stress. Therefore, it is important to take care of their physical and mental health by improving access to community and home-based play spaces, therapy, health and psychosocial services, as well as other service navigation support.

Creatively delivering parenting education focused on enhancing caregiver capacities to become more responsive, promote maternal and child health and wellbeing, as well as adequate nutrition services will be at the core of driving nurturing care, and hence improving ECD outcomes. Critical and essential health and nutrition, as well as other social services, can be delivered and sustained during the pandemic period with adherence to adequate infection prevention and control measures. Furthermore, health education and promotion in addition to continuous engagement and referrals of caregivers and families through community health structures is crucial.

Innovative culturally acceptable strategies that transcend the existing pandemic barriers with a strong emphasis on strengthening community-based reproductive, maternal, newborn, child and adolescent health services are required. These services include family planning services; maternal nutrition such as promoting the uptake of iron and folic acid supplements; antenatal care; seeking skilled delivery and post-natal services; as well as essential nutrition services to support infant and young child feeding, routine growth monitoring and counselling through baby-friendly community initiatives. These should be further complemented with adequate transport to a health facility during curfew or lockdown situations. In some settings, health services are being taken to families in remote locations, especially those through mobile vans or clinics, thereby enabling greater and equitable nurturing care support for the youngest of children during the COVID-19 crisis [101,102]. Essential new-born care should be an area of sustained focus: early initiation and assessment for exclusive breastfeeding, addressing danger signs for referrals and timely linkages to health services. Mothers and caregivers also need timely referrals and access to services for treatment of maternal and child undernutrition. Integrated community case management of common childhood diseases, in particular malaria, diarrhea, pneumonia and malnutrition, should not be neglected.

Mental health, often overlooked, has risen to the consciousness of policy makers and donors. This pandemic provides an opportunity to take the innovations and expand mental health/psycho-social services throughout Kenya. All people, young and old, are facing mental and emotional difficulties. Caregivers juggling full-time jobs, caring for others such as the elderly and children at home, are feeling especially overwhelmed. Development agencies are supporting families’ mental health and psycho-social wellbeing where possible by establishing phone helplines to increase access to free professional mental health support. Referral systems are also being established through these helplines, and this needs to be expanded and accessible. Simple tips and exercises, relevant to both the young and old, are being broadcast on TV, radio, social media (Facebook (Facebook, Inc., San Francisco, CA, USA) or WhatsApp (WhatsApp Inc., San Francisco, CA, USA)) and through short videos.

The design and utilization of mHealth can lead to improved ECD outcomes. In particular, the use of telehealth consultations where possible, with health professionals, can also help to minimize hospital visits. Efforts should be made to scale up provision of nurturing care through integration into their health systems by adopting the mNurturingCare app in clinical encounters and at the community level [103]. In addition, partnerships with the local communities are important to increase engagement and dialogue on the measures for supporting nurturing through heightened communication with health professionals who can offer prompt identification of complications and provide appropriate referrals.

### 6.2. Economic Support

As COVID-19 is negatively affecting the economic situation of many families in Kenya, especially those who work in the informal sector and/or were already in precarious economic situations prior to the outbreak, innovative approaches such as cash transfers are necessary. In various emergencies around the world and in Kenya, conditional and unconditional cash transfer programs have provided an economic safety net and positively impacted health during difficult periods such as the one we face [104,105,106]. Although the government has instituted social protection schemes in the form of cash transfers to cushion vulnerable families, the need is greater. More investments are required, particularly programs that support those who were previously working in the informal sector and have lost incomes. This can be achieved through reallocation of funds to social protection to increase the resources available for cash transfers and food supplies in order to mitigate socio-economic impacts, including addressing food insecurity [58,66]. These measures should be accompanied with the introduction of functional community mechanisms for identifying vulnerable households and children who lack access to basics such as food, water, shelter and healthcare, and provide targeted support. Social safety nets for vulnerable families during this pandemic enable them to provide nurturing care to promote resilience among children, despite the stressors surrounding them.

### 6.3. Social and Child Protection Support

It is clear that many social and child protection services targeted at children and adolescents in Kenya, such as meals and sanitary pads among others, have largely been delivered through childcare and school platforms; with school closures, this avenue is not viable. There is a need to build and leverage community level programs and policy support, targeting children at risk of abuse and neglect, and adolescents at risk of early pregnancy, to ameliorate the negative effects of the pandemic, such as poor nurturing care environments and transactional sex for food and pads among adolescent girls, all of which subsequently leading to poor child development outcomes. Mitigation strategies should focus on safety nets for poor families in addition to identifying and supporting vulnerable children and adolescents within these families. Urgent strategies are required to protect young children and girls from the increased gender-based violence during the pandemic period. These strategies include improved access to psychosocial support services through community agents or call-in centers to reduce caregiver stress, expansion of social and child protection services such as family tracing and reunification of separated or orphaned children, and increased delivery of reproductive health services through mobile reproductive health services and telemedicine in remote communities.

For orphaned children, evidence indicates that family-based care is better for young children than institutional care [80]. The landmark longitudinal study of Romania’s orphans led by the Bucharest Early Intervention Project showed that brain development can be severely affected when orphaned children are in institutional care without nurturing care [107]. The study indicates that this effect can last over a person’s lifetime. Key stakeholders therefore should find safe and protective homes for orphaned children with other relatives and ensure they benefit from social and child protection services. This could also include conditional and unconditional cash transfers to help relatives of orphaned children that want to take care of them in supportive family environments.

Other critical community outreach strategies are also required to address poor nurturing care environments and rising transactional sex in partnership with nurses and community workers. These include expanding community outreach for nurturing care skills among caregivers and increasing access to sexual and reproductive health and rights education, as well as distribution of sanitary pads and contraceptives among adolescent girls. These strategies must be accompanied by facility and community-based youth-friendly reproductive health services. Special attention needs to be given to the children of young teen mothers through social protection schemes to ensure that their children can achieve the highest developmental potential during this period.

### 6.4. Child Development and Learning Support

Children and their primary caregivers/parents need social and educational support to ensure socio-emotional well-being, safety and security from violence and harm and opportunities to boost young children’s brain development. Children are separated from peers and extended family members such as grandparents, and are unable to attend early learning centers, daycares, and crowded areas. Some children, just by being home, are exposed to domestic violence; early evidence points to an increase in such cases as stress levels of families’ increase [72]. Concerted efforts including awareness raising are necessary to reduce violence against children. Some innovations are being tested in this area by development agencies. For instance, the use of TV, radio, pre-made videos and social media to support early childhood educators, teachers, and family members with simple ways to support young children’s learning and development at home has been observed.

Parents and caregivers are the most important support structure for young children, and their ability to nurture adequately while remaining physically and mentally healthy is critical. Parents and caregivers should therefore become a critical target audience for ECD stakeholders in Kenya and the region, ensuring that nurturing care becomes a family-centered with a whole-society approach. With the additional burdens being placed on parents and caregivers, they need to prioritize their physical and mental health. The strategies suggested above in the section on health and health systems support can be crucial. Additionally, parents and caregivers also need practical tools and guidance to enable them to provide early nurturing care in the home environment, particularly providing opportunities for early learning as well as increasing community-led sanitation and nutrition programs. This includes strategies on how to regularly interact and communicate with children and provide them with age-appropriate play and learning resources at home, using locally available, low-cost or household materials easily found in their surrounding environment. The Care for Child Development package is a useful intervention that can be cascaded widely, leveraging multi-sectoral community counselling platforms to encourage greater parental or caregiver responsiveness through communication and play [108]. This will contribute to the healthy development and growth of children by supporting caregivers to build stronger relationships with their children and solve challenges in providing nurturing care.

For those children who may have been attending babycares that have been closed due to the crisis, the caregivers need practical support to provide quality early learning in the home environment. Linkages with child and social protection actors should be strengthened to promote nurturing care, as parents have now taken over childcare and schooling in the home while balancing dual roles of work and managing the household. In the critical early years, young children need at least one loving and trusted adult to feel secure, grow and develop holistically. In this period of adversity, children need nurturing relationships with caregivers and families to provide a buffer to counterbalance the hardships [10].

## 7. Conclusions

COVID-19 is still ravaging Kenya and most of the world. There is still a lot to learn about what can work and what cannot. Little research is currently published on how to support nurturing care for children under 4 years in the wake of such a pandemic, especially in Africa. While the impacts of the pandemic on the lives of young Kenyan children and their families have been severe, and evidence around further impacts is coming to light, there are opportunities to learn and “build back better”. Interventions need to address five critical areas: direct health impacts, health and nutrition systems impacts, economic impacts, social and child protection impacts and child development learning impacts. There is need to leverage technology and use a community-based approach to support continuity of nurturing care services with timely referral and follow-up to a wide range of cross-sectoral services including psychosocial support. Kenya has an opportunity to learn from other countries about how to develop practical and feasible guidance to reopen childcare centers and early learning spaces, given the low incidence of COVID-19 in younger children. Tailored plans are required for children with unique needs, and consideration should be given to provide holistic and inclusive support. The government and development actors need to consider increasing their investments to scale-up nurturing care through the development of policies and coordinated intervention programs during this pandemic period. This paper took stock of what we currently know about the impacts of COVID-19 on nurturing care for the youngest Kenyans, but this is just the tip of the iceberg. Although our review paper has provided useful insights and made an important contribution to the body of knowledge, the key limitation is that it is mainly based on secondary sources including grey literature, and it did not rely on primary data. Further primary research and investigation on the youngest children and the ability of children’s ecology to provide nurturing care is needed, as is the further testing of new ideas. Primary research is required particularly to explore the mental health impacts on caregivers and children; understand how domestic violence has impacted nurturing care; and shed more light on the experiences and support available to teenage mothers and caregivers of children living with disabilities. Additional evidence would light the way forward for Kenya and similar settings to ensure its youngest citizens can reach their full developmental potential.

## Figures and Tables

**Figure 1 ijerph-17-07028-f001:**
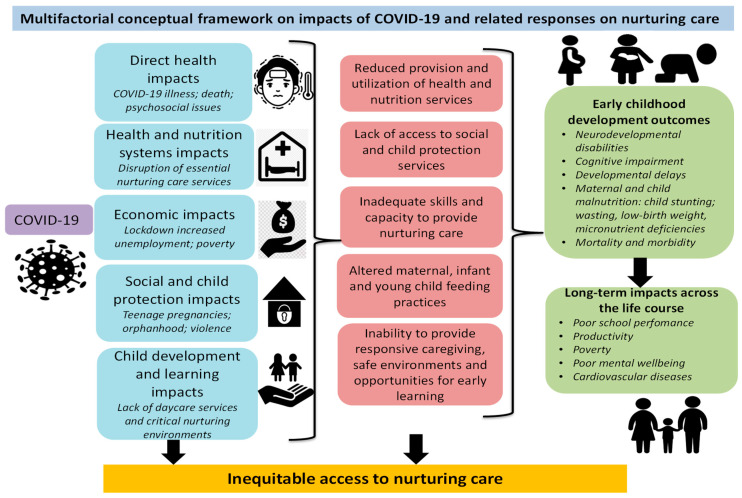
Authors’ conceptualization of the impacts of Coronavirus disease 2019 (COVID-19) on nurturing care.

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
