# Peer review of "Reorienting Nurturing Care for Early Childhood Development during the COVID-19 Pandemic in Kenya: A Review"

_ijerph, 2020, doi:10.3390/ijerph17197028_

Round 1

Reviewer 1 Report

A thoughtful and apt manuscript on the impact of Covid-19 on early childhood development in Kenya. The authors have explored how the pandemic is affecting Kenyan society and early childhood development.

In terms of providing support to parents, could the UNICEF Care for Child Development package be used to encourage greater parental responsiveness through communication and play?

Although there is a strong sense of not knowing what will happen next, I feel the authors could give some foresight into how a 'build-back-better' model could work in Kenya. This could be in response to the low incidence rates of Covid 19 in younger children and how Kenya could learn from other countries about the need to re-open child centres and schools particularly for very vulnerable children and those with disabilities.

Some recommendations on how to best support children with developmental delay and disability in communities during the lockdown would be helpful.

Author Response

1. A thoughtful and apt manuscript on the impact of Covid-19 on early childhood development in Kenya. The authors have explored how the pandemic is affecting Kenyan society and early childhood development.

Thank you

2. In terms of providing support to parents, could the UNICEF Care for Child Development package be used to encourage greater parental responsiveness through communication and play?

Yes, this is a good suggestion and we have added it as follows [Line 535-540]: The Care for Child Development package is a useful intervention that can be cascaded widely, leveraging multi-sectoral community counselling platforms to encourage greater parental or caregiver responsiveness through communication and play [108]. This will contribute to the healthy development and growth of children by supporting caregivers to build stronger relationships with their children and solve challenges in providing nurturing care.

3. Although there is a strong sense of not knowing what will happen next, I feel the authors could give some foresight into how a 'build-back-better' model could work in Kenya. This could be in response to the low incidence rates of Covid 19 in younger children and how Kenya could learn from other countries about the need to re-open child centres and schools particularly for very vulnerable children and those with disabilities.

This has been expanded as follows [Line 555-583]: Interventions for immediate consideration include those that leverage technology to provide comprehensive information, education and communication, and address unmet needs including general health, therapy and psychosocial support. Further, there is need to use a community-based approach to support continuity of nurturing care services with timely referral and follow-up to a wide range of cross-sectoral services while addressing financial insecurity through provision of basic needs such as food and water. Kenya has an opportunity to learn from other countries about how to re-open childcare centers particularly for very vulnerable children and those with disabilities given the low incidence of COVID-19 in younger children. The country will need to develop practical and feasible guidance and strategies, that can be revised and adapted depending on the transmission dynamics in communities. Families of children with disabilities will need to be supported families using a shared decision-making approach by the health, social and child protection services. Tailored plans are required for children with unique needs and consideration should be given to provide holistic support and this should be communicated in ways that promote inclusion. The government and development actors need to consider increasing their investments to scale-up nurturing care in care during this pandemic period. This includes development of policies and coordinated intervention programs for nurturing care to support children in their most formative years in both pandemic and non-pandemic times in LMICs. Programs should be bolstered with relevant monitoring and evaluation frameworks to keep track of progress and outcomes. This paper took stock of what we currently know about the impacts of COVID-19 on nurturing care for the youngest Kenyans, but it is just the tip of the iceberg. Although our review paper has provided useful insights and made an important contribution to the body of knowledge, the main limitation is that it is mainly based on secondary sources including grey literature and did not rely on primary data. Further research and investigation on the youngest children and the ability of children’s ecology to provide nurturing care is needed as is further testing of new ideas. Additional evidence could light the way forward for Kenya and similar settings, to ensure its youngest citizens can reach their full developmental potential. Further, primary research is required to explore the mental health impacts on caregivers; understand how domestic violence has impacted nurturing care, and shed more light on the experiences and supports available to teenage mothers and caregivers of children living with disabilities.

 4. Some recommendations on how to best support children with developmental delay and disability in communities during the lockdown would be helpful.

We have added as follows [Line 418-423]: Children with disabilities may experience stress, have underlying health conditions that increase their risk of complications from COVID-19, and may also be unable to access therapy during this period. Similarly, their caregivers may have heightened stress. Therefore, it is important to take care of their physical and mental health by improving access to community and home-based play spaces, therapy, health and psychosocial services as well as other service navigation support.

Reviewer 2 Report

Thank you for the opportunity to review the manuscript “Re-orienting nurturing care for early childhood development during the COVID-19 pandemic in Kenya: A review”.

The manuscript is highly interesting because it provides insight into the situation of early childhood in Kenya after the COVID-19 pandemic. It is an enjoyable text to read, well structured and very interesting. However its scientific content is not too deep since it does not delve too deeply into socio-sanitary aspects of the situation in Kenya, but this is not a problem from my point of view for it to be accepted in the journal. Also as a limitation given the type of work, it should be noted the excessive number of authors who sign the manuscript. It might be appropriate to indicate the specific contribution of each of them to the work.

Apart from the above, some comments below can help improve the work.

- On page 2, (lines 50-54), the objectives of the work are exposed. I think they should be written in a more specific and detailed way, with verbs in the infinitive.

- The conclusions section is too poor and should be expanded. It could be developed by commenting on the main results, offering certain opinions of the authors themselves on the situation in Kenya as well as in relation to previous studies. The authors should comment on limitations of the study that they may see. Finally in this section, it would be interesting to offer suggestions and lines of research for future studies that other authors or themselves could develop, especially from an empirical perspective.

- The references present some errors according to the journal's rules, so they should be reviewed.

Given that the approach and the contents of the manuscript are of great interest as it allows an understanding of the reality of early childhood in Kenya, and despite a certain informative nature and not too much scientific in-depth on the subject, my opinion is that the article can be accepted for publication in the journal, after slight changes and modifications.

Finally, I want to thank again for having the opportunity to read the work and I encourage the authors to continue working in this line.

Author Response

1. Thank you for the opportunity to review the manuscript “Re-orienting nurturing care for early childhood development during the COVID-19 pandemic in Kenya: A review”.

The manuscript is highly interesting because it provides insight into the situation of early childhood in Kenya after the COVID-19 pandemic. It is an enjoyable text to read, well-structured and very interesting. However its scientific content is not too deep since it does not delve too deeply into socio-sanitary aspects of the situation in Kenya, but this is not a problem from my point of view for it to be accepted in the journal.

Thank you. Indeed the socio-sanitary aspects are not too deep as we wanted to strike the right balance for a variety of audiences. Due to the multi-faceted nature of the impacts, we think that further analysis may be needed in other separate work.

2. Also as a limitation given the type of work, it should be noted the excessive number of authors who sign the manuscript. It might be appropriate to indicate the specific contribution of each of them to the work.

Thank you. We will indicate their contributions. Our team was multi-disciplinary and we worked extensively on this piece over several months with contributions from the various authors, and held internal meetings where we critically reviewed the sum of our contributions.

Apart from the above, some comments below can help improve the work.

3. On page 2, (lines 50-54), the objectives of the work are exposed. I think they should be written in a more specific and detailed way, with verbs in the infinitive.

This has been edited as follows [Line 50-54]: ‘Through a conceptual framework, the objective of this paper is to elucidate how nurturing care for the youngest Kenyans is being threatened. We also present our reflections on how to re-orient and support nurturing care during and after the pandemic period. Further, we describe a myriad of measures and strategies that key stakeholders in Kenya can adopt to reduce these threats to young children’s ability to survive, thrive and transform their societies.’

4. The conclusions section is too poor and should be expanded. It could be developed by commenting on the main results, offering certain opinions of the authors themselves on the situation in Kenya as well as in relation to previous studies. The authors should comment on limitations of the study that they may see. Finally in this section, it would be interesting to offer suggestions and lines of research for future studies that other authors or themselves could develop, especially from an empirical perspective.

Thank you. This has been expanded as follows [Line 555-583]: Interventions for immediate consideration include those that leverage technology to provide comprehensive information, education and communication, and address unmet needs including general health, therapy and psychosocial support. Further, there is need to use a community-based approach to support continuity of nurturing care services with timely referral and follow-up to a wide range of cross-sectoral services while addressing financial insecurity through provision of basic needs such as food and water. Kenya has an opportunity to learn from other countries about how to re-open childcare centers particularly for very vulnerable children and those with disabilities given the low incidence of COVID-19 in younger children. The country will need to develop practical and feasible guidance and strategies, that can be revised and adapted depending on the transmission dynamics in communities. Families of children with disabilities will need to be supported families using a shared decision-making approach by the health, social and child protection services. Tailored plans are required for children with unique needs and consideration should be given to provide holistic support and this should be communicated in ways that promote inclusion. The government and development actors need to consider increasing their investments to scale-up nurturing care in care during this pandemic period. This includes development of policies and coordinated intervention programs for nurturing care to support children in their most formative years in both pandemic and non-pandemic times in LMICs. Programs should be bolstered with relevant monitoring and evaluation frameworks to keep track of progress and outcomes. This paper took stock of what we currently know about the impacts of COVID-19 on nurturing care for the youngest Kenyans, but it is just the tip of the iceberg. Although our review paper has provided useful insights and made an important contribution to the body of knowledge, the main limitation is that it is mainly based on secondary sources including grey literature and did not rely on primary data. Further research and investigation on the youngest children and the ability of children’s ecology to provide nurturing care is needed as is further testing of new ideas. Additional evidence could light the way forward for Kenya and similar settings, to ensure its youngest citizens can reach their full developmental potential. Further, primary research is required to explore the mental health impacts on caregivers; understand how domestic violence has impacted nurturing care, and shed more light on the experiences and supports available to teenage mothers and caregivers of children living with disabilities.

5. The references present some errors according to the journal's rules, so they should be reviewed.

Thank you we have reviewed these thoroughly and present these in the Harvard format. All the changes are tracked.

6. Given that the approach and the contents of the manuscript are of great interest as it allows an understanding of the reality of early childhood in Kenya, and despite a certain informative nature and not too much scientific in-depth on the subject, my opinion is that the article can be accepted for publication in the journal, after slight changes and modifications.

Thank you for your feedback. The authors do appreciate it.

7. Finally, I want to thank again for having the opportunity to read the work and I encourage the authors to continue working in this line.

Thank you